# Haploinsufficiency and Alzheimer’s Disease: The Possible Pathogenic and Protective Genetic Factors

**DOI:** 10.3390/ijms252211959

**Published:** 2024-11-07

**Authors:** Eva Bagyinszky, Seong Soo A. An

**Affiliations:** 1Department of Industrial and Environmental Engineering, Graduate School of Environment, Gachon University, Seongnam 13120, Republic of Korea; 2Department of Bionano Technology, Gachon Medical Research Institute, Gachon University, Seongnam 13120, Republic of Korea

**Keywords:** Alzheimer’s disease, haploinsufficiency, loss of function, animal models, neurodegeneration, neuroprotection

## Abstract

Alzheimer’s disease (AD) is a complex neurodegenerative disorder influenced by various genetic factors. In addition to the well-established amyloid precursor protein (*APP*), Presenilin-1 (*PSEN1*), Presenilin-2 (*PSEN2*), and apolipoprotein E (*APOE*), several other genes such as Sortilin-related receptor 1 (*SORL1*), Phospholipid-transporting ATPase ABCA7 (*ABCA7*), Triggering Receptor Expressed on Myeloid Cells 2 (*TREM2*), Phosphatidylinositol-binding clathrin assembly protein (*PICALM*), and clusterin (*CLU*) were implicated. These genes contribute to neurodegeneration through both gain-of-function and loss-of-function mechanisms. While it was traditionally thought that heterozygosity in autosomal recessive mutations does not lead to disease, haploinsufficiency was linked to several conditions, including cancer, autism, and intellectual disabilities, indicating that a single functional gene copy may be insufficient for normal cellular functions. In AD, the haploinsufficiency of genes such as *ABCA7* and *SORL1* may play significant yet under-explored roles. Paradoxically, heterozygous knockouts of *PSEN1* or *PSEN2* can impair synaptic plasticity and alter the expression of genes involved in oxidative phosphorylation and cell adhesion. Animal studies examining haploinsufficient AD risk genes, such as vacuolar protein sorting-associated protein 35 (*VPS35*), sirtuin-3 (*SIRT3*), and *PICALM*, have shown that their knockout can exacerbate neurodegenerative processes by promoting amyloid production, accumulation, and inflammation. Conversely, haploinsufficiency in *APOE*, beta-secretase 1 (*BACE1*), and transmembrane protein 59 (*TMEM59*) was reported to confer neuroprotection by potentially slowing amyloid deposition and reducing microglial activation. Given its implications for other neurodegenerative diseases, the role of haploinsufficiency in AD requires further exploration. Modeling the mechanisms of gene knockout and monitoring their expression patterns is a promising approach to uncover AD-related pathways. However, challenges such as identifying susceptible genes, gene–environment interactions, phenotypic variability, and biomarker analysis must be addressed. Enhancing model systems through humanized animal or cell models, utilizing advanced research technologies, and integrating multi-omics data will be crucial for understanding disease pathways and developing new therapeutic strategies.

## 1. Introduction

Haploinsufficiency is a genetic condition in which one copy of a gene is non-functional or deleted, resulting in insufficient gene, mRNA, and protein function. Various mutations, including large deletions, copy number variants, frameshift mutations, stop codon mutations, or even missense mutations, can lead to a reduced expressions of mRNA and protein [1,2,3]. Several genes associated with haploinsufficiency play crucial roles in development, the regulation of cellular functions, and the maintenance of homeostasis [2]. Initially, haploinsufficiency was investigated in developmental disorders, autism, epilepsy, major depressive disorders [4,5], and cancers [6,7,8]. The significance of haploinsufficiency in neurodegenerative diseases is being investigated, particularly with a reduced expressions of progranulin (*GRN*) in patients with frontotemporal dementia (FTD) [9,10]. Similarly, repeat expansions in the *C9orf72* gene can be associated with both gain-of-function and loss-of-function mechanisms in FTD [11].

It is possible that haploinsufficiency could contribute to the pathogenesis of Alzheimer’s disease (AD), as many genetic, age-associated, and environmental factors could influence the complex pathogenicity of AD. While majority of AD occur after the age of 60 without a clear genetic cause, while approximately 5–10% of AD cases are linked to familial genetic factors, such as mutations in the amyloid precursor protein (*APP*), Presenilin-1 (*PSEN1*), or Presenilin-2 (*PSEN2*) genes. Mutations in these genes are inherited in an autosomal dominant pattern; however, *APP* Val673Ala was shown to be inherited in an autosomal recessive pattern [12,13,14].

Late-onset AD (LOAD), which occured in individuals over 65 years of age, presented with a more complex disease etiology, as environmental, lifestyle, and genetic factors all contribute to disease progressions. The strongest genetic risk factor for LOAD was the apolipoprotein E (*APOE*) gene with three main alleles: E2, E3, and E4. The majority of individuals carried the E3 allele, while the E2 allele was found to be protective against neurodegeneration among minor populations. Homozygous or heterozygous forms of the E4 allele were associated with a higher risk of AD progression through diverse mechanisms, such as reduced amyloid clearance, increased Tau phosphorylation, abnormal mitochondrial function, and abnormal lipid metabolism [15]. Other than *APOE* E4, no specific causative genetic factor was definitively linked to LOAD, although several risk factors were suggested. Additionally, environmental factors and gene–environment interactions could influence in causing LOAD [15,16].

As genome-wide association studies (GWAS) and next-generation sequencing technologies became more accessible, novel genetic factors impacting AD risk were identified [16]. Some of the genetic factors were directly involved in amyloid processing (e.g., *APP*; *PSEN1*; *PSEN2*; ADAM metallopeptidase domain 10; *ADAM10*; beta-secretase 1; *BACE1*), while others affected amyloid metabolism indirectly through pathways, such as amyloid transport, degradation, or clearance (e.g., ATP-binding cassette subfamily A member 7, *ABCA7*; Phosphatidylinositol-binding clathrin assembly protein, *PICALM*; APOE; clusterin, *CLU*; Sortilin-related receptor 1, *SORL1*). Additionally, AD risk genes could influence alternative pathways, including inflammation, autophagy, metabolism, and signal transduction (e.g., calcium signaling, protein kinase C signaling, Wnt signaling, Notch signaling). Mutations in these genes contributed to AD through both gain-of-function and loss-of-function mechanisms. In addition the reduced gene expression, caused by frameshift, stop codon, or missense mutations could play a significant role in AD onset [16,17].

Besides the pathogenic or risk variants, several neuroprotective factors were discovered in AD. As mentioned earlier, the most notible protective variant was the *APOE* E2 allele. However, additional protective variants were discovered; more notably, the Christchurch variant in *APOE* (Arg136Ser) was found to prevent or slow down the Tau pathology, inflammation, or neurodegeneration [18,19]. Additionally, an *APP* variant, Ala673Thr, was found to protect against AD-related neurodegeneration and disease-related alterations of biomarkers in cerebrospinal fluid (CSF) or blood [20]. Furthermore, variants in other genes, including *ABCA7* (Gly215Ser, Val1613Met) or fibronectin-1 (*FN1*), were found to protect against AD progression [21,22,23].

Several attempts were performed to develop drugs against AD. A majority of the available drugs (including acetylcholine inhibitors) are focusing on reducing the disease symptoms; however, they cannot prevent the disease progression. Two drug candidates, aducanumab and lecanemab, showed disease-modifying effects in clinical trials; however, further studies are needed to verify their long-term effects and safety [24,25,26]. The development of gene therapy against neurodegenerative diseases, including AD, may also be promising for patients [27,28].

This review discusses the potential role of haploinsufficiency in AD, considering evidence from cell and animal models (e.g., 5xFAD) suggesting that heterozygous knockout of certain genes contributes to AD pathogenesis. Interestingly, heterozygous knockout of certain genes, such as *APOE*, revealed protective effects against AD progression. This manuscript highlights genetic factors that either contribute to AD pathogenesis or provide neuroprotection through haploinsufficiency.

## 2. *PSEN1* and *PSEN2* and the Possibility of Haploinsufficiency

The majority of mutations in *PSEN1* or *PSEN2* were associated with gain-of-function mechanisms, leading to upregulated gamma-secretase activity. However, loss-of-function mutations in *PSEN1* were also reported, such as *PSEN1* Arg352 duplication, *PSEN1* Ala246Glu, *PSEN1* Cys410Tyr, and *PSEN1* Leu435Phe. It remains unclear whether these mutations act through reduced *PSEN1* expression, but they were found to inhibit gamma-secretase activity. Multiple studies revealed that Ala246Glu increases the Aβ42/Aβ40 ratio, although both Aβ42 and Aβ40 levels are reduced in the mutant cell lines [29,30]. Ala246Glu was also found to inhibit neuronal development and reduce neuronal survival by impairing Wnt and Notch signaling [31]. Furthermore, *PSEN1* Ala246Glu was associated with additional impairments, including abnormal autophagy, mitophagy, mitochondrial dysfunction, and elevated calcium release from the endoplasmic reticulum [32,33]. Cell models of Arg352 duplication revealed an increased Aβ42/Aβ40 ratio, though both Aβ42 and Aβ40 levels were reduced in cultured cells [34]. A more recent study on *PSEN1* Arg352 duplication showed that the mutation reduced Aβ42 levels and abolished Aβ40 production [29]. *PSEN1* Cys410Tyr was associated with reduced *PSEN1* expression in mouse and human non-neuronal and neuronal-like cells [35]. Cell studies showed that Cys410Tyr resulted in a reduction of both long and short amyloid levels, and the ratio of short amyloid to long amyloid (37 + 38 + 40)/(42 + 43) was lower, while the Aβ42/Aβ40 ratio was elevated [36]. This mutation also resulted in abnormal Notch1 and neurexin processing [35,37]. *PSEN1* Leu435Phe was found to disrupt both APP and Notch processing. The mutation was associated with reduced levels of Aβ42, Aβ40, and C-terminal APP. It was also suggested that Leu435Phe could impair synaptic plasticity in the hippocampus [38,39]. However, mouse models revealed that this mutation increased Aβ43 levels in plaques [40].

Taken together, loss-of-function mutations in *PSEN* genes resulted in decreased gamma-secretase activity and altered amyloid processing. However, they still increased the Aβ42/Aβ40 ratio through a partial or complete loss of short amyloid generation. These loss-of-function mutations were found to cause neurodegeneration through amyloid-independent mechanisms, including Tau phosphorylation, reduced synaptic plasticity, impaired protein trafficking, apoptosis, and calcium dysregulation [40,41,42,43,44].

A few reported stop codon and frameshift mutations were observed in *PSEN1* and *PSEN2*, which were associated with diverse clinical phenotypes (Table 1). Haploinsufficiency was proven in the case of *PSEN1* Pro242Leufs, which resulted in acne inversa [45,46]. Haploinsufficiency was also possible with *PSEN1* Trp294Ter in a patient with retinitis pigmentosa and acute encephalopathy [47], *PSEN1* Ser357Ter in a patient with cognitive decline and cerebral amyloid angiopathy (CAA) [48], and *PSEN1* Gly378fs in a Moroccan family with AD [49]. A cell study on the Gly378fs mutation in N2A cells indicated reduced expression of both Aβ42 and Aβ40, with an elevated Aβ42/Aβ40 ratio. This mutation was initially classified as a variant of unknown significance (VUS), but after in vitro studies, it was reclassified as a probable pathogenic variant through loss-of-function mechanisms. However, no studies were investigated whether this mutation could disturb the amyloid-independent pathways [50].

In *PSEN2*, multiple frameshift mutations were identified, including Lys82fs [51], Lys115Glufs10 [43,52], Glu126fs [49], Lys306fs [49], Gly359Leufs74 [51], and Ala394Profs*8 [53]. In the case of *PSEN2* Glu82fs, reduced levels of *PSEN2* transcript were found in the frontal cortex and hippocampal regions of the patient’s brain [51]. The truncated *PSEN2* protein resulting from the Lys115Glufs*10 mutation was detected in the patient’s biological fluids, and reduced wild-type transcript levels were observed compared to controls. Cell models of this mutation in fibroblasts also revealed reduced *PSEN2* transcript and protein levels [52]. The *PSEN2* Gly359Leufs*74 mutation in patient-derived lymphoblasts was suggested to present reduced stability and concentrations of both protein and transcript levels [51]. A stop-gain variant, *PSEN2* Gly117Ter, was also reported, although no functional studies were conducted on it [54].

**Table 1 ijms-25-11959-t001:** Mutations in *PSEN1* and *PSEN2*, which were proven to be related or possibly associated with haploinsufficiency. Variants, which were associated with reduced *PSEN* transcript or protein levels are highlighted with bold.

	Mutation	Disease	AOO	Family History	Amyloid Changes	Transcript/Protein Changes	References
** *PSEN1* **	**Pro242fs**	**Acne inversa**	**Young onset**	**Positive**	**No effect on amyloid cleavage, but altered Notch signaling**	**Reduced *PSEN1* mRNA and protein**	[45,46]
	Trp294Ter	Acute Encephalopathy, Retinitis Pigmentosa	13 years	Probable positive	NA	NA	[47]
	Ser357Ter	Cognitive decline, CAA	55 years	NA	NA	NA	[48]
	Gly378fs	AD, aphasia	63	Probable positive	Reduced Ab42 and Ab40, Increased Ab42/40 ratio	NA	[49,50]
*PSEN2*	**Lys82Ilefs*27**	**FTD**	**NA**	**NA**	**NA**	**Reduced *PSEN2* transcript in hippocampus and frontal lobe**	[51]
	**Lys115Glufs*10**	**AD, language impairment**	**Late 40s–50s**	**Probable positive**	**Reduced Ab40 levels, Ab42 and 38 failed to be detected**	**Reduced *PSEN2* transcript and protein**	[43,52]
	Gly117Ter	AD	55	Probable positive	NA	NA	[54]
	Glu126fs	AD	60	Probable positive	Slightly reduced Ab42/40	NA	[50]
	Lys306fs	AD, aphasia	55	Probable positive	NA	NA	[49]
	**Gly359Leufs*74**	**Multi-domain MCI to AD**	**55**	**NA**	**NA**	**Reduced *PSEN2* mRNA and protein**	[51]
	Gly359Leufs*74	ALS	NA	NA	NA	NA	[51]
	Ala394Profs*8	AD	68	Probable positive	NA	NA	[53]

These findings suggest that haploinsufficiency in *PSEN1* and *PSEN2* should not be overlooked. In addition to AD, *PSEN1* or *PSEN2* haploinsufficiency may be related to other phenotypes, such as acne inversa [45,46], retinitis pigmentosa [24], or FTD [51]. Mutations associated with haploinsufficiency altered amyloid metabolism through loss-of-function mechanisms and reduced short amyloid production, as seen with the *PSEN1* Gly378fs mutation, which was associated with an elevated Aβ42/Aβ40 ratio [50]. Reduced *PSEN* transcript and protein levels were observed in patients with *PSEN1* Pro242fs, *PSEN2* Lys115Glufs*10, and *PSEN2* Gly359Leufs*74 mutations. However, further studies are needed to understand the causal relationship between reduced *PSEN* transcripts and pathogenic mechanisms [23]. The *PSEN2* Lys115Glufs*10 mutation was associated with reduced gamma-secretase activity and APP cleavage and impaired Notch cleavage. Alternatively spliced *PSEN2* products were detected in the brain tissue of patients with this mutation, potentially contributing to neurodegeneration through loss-of-function mechanisms [52].

In addition to the classical amyloid cleavage-related pathways, *PSEN* haploinsufficiency can induce amyloid-independent mechanisms, such as Notch signaling, Tau phosphorylation, and abnormal cellular functions [52]. Zebrafish models with a *PSEN2* frameshift mutation (Asn140fs, homologous to Asn141 in humans) revealed reduced gamma-secretase activity in mutant zebrafish. This frameshift mutation led to decreased *PSEN2* expression in the brain, likely due to nonsense-mediated decay. Gene expression analysis in zebrafish suggested that the *PSEN2* frameshift mutation altered the expression of genes associated with oxidative phosphorylation, ribosomal function, MAPK signaling, Notch signaling, adhesion, and extracellular matrix interactions. This study suggested that frameshift or stop codon mutations in *PSEN2* induce cellular stress, contributing to AD pathology [55].

The presenilin hypothesis of AD revealed that reduced or absent *PSEN* expression can also be implicated in neurodegenerative pathways. Knocking out *PSEN* genes in mice resulted in memory dysfunction and reduced synaptic plasticity. In these mice, reduced gene expression of NMDA receptor-mediated synaptic responses and cAMP-response element (CRE)-dependent pathways led to gliosis and elevated levels of phosphorylated Tau [56,57]. Figure 1 summarizes the gain-of-function and loss-of-function mechanisms in disease-related pathways involving *PSEN* genes.

## 3. Haploinsufficiency of *SORL1* in AD

*SORL1* was identified as a risk factor for both EOAD and LOAD. The SORL1 protein regulates protein transport between the endosomes and the trans-Golgi network. By transporting APP and determining its fate, SORL1 can influence the amyloid processing [58,59,60,61]. Several frameshifts and stop codon mutations were described in *SORL1*, though only a few were proven to impact amyloid metabolism through haploinsufficiency (Table 2).

Quantitative PCR on lymphoblasts from a patient with the *SORL1* Gly447Argfs*22 mutation revealed reduced *SORL1* expression in the patient-derived lymphoblasts. The reduced *SORL1* expression significantly increased amyloid deposition and decreased its clearance [62]. Similarly, human-derived microglial stem cells with the *SORL1* Arg744Ter mutation showed abolished *SORL1* expression and deficits in amyloid uptake by the microglia [63]. A study of the His962Profs*45 mutation in human-induced pluripotent stem cells (iPSCs) from AD patients revealed reduced SORL1 protein levels in carriers. In these patients, the number of larger early endosomes increased compared to controls, which resulted in impaired endocytosis [64]. Cells derived from patients with the *SORL1* Cys1431Trpfs*2 mutation exhibited elevated endosomal APP deposition. Furthermore, this mutation reduced synaptic density and synapsin-I levels in the neurons [65,66]. Patient-derived blood cells with the *SORL1* Cys1478Ter mutation showed reduced mRNA expression due to truncation, but there was no information available on whether this mutation affects amyloid metabolism or other neurodegenerative pathways [66].

Zebrafish models were generated to model the human EOAD-like phenotypes. Since Cys1481 in zebrafish was conserved with human Cys1478, the goal was to model the potential effects of the *SORL1* Cys1478Ter mutation. The mutation resulted in a significant reduction in normal *SORL1* expression, and abnormal transcripts were also detected. The *SORL1*-truncation mutation was associated with alterations in the expression of genes involved in energy production, mRNA translation, and mTORC1 signaling, suggesting that this zebrafish model may be useful for presenting cellular stress associated with early disease stages [67]. *SORL1* Trp1821Ter was also generated in a zebrafish model. Quantifying the RNA levels by qPCR revealed reduced levels of normal *SORL1* mRNA. Furthermore, the wild-type protein levels were also reduced in the mutant zebrafish. Analyzing the differentially expressed genes in terms of *SORL1* Trp1821Ter revealed alterations in the expression of genes involved in ribosomal function (e.g., hepatocyte nuclear factor 4alpha, Hnf4a) or mitochondrial functions (e.g., Vdac1). However, this study failed to find any proof that the mutation could result in alterations in the expression of genes involved in endo-lysosomal functions [68].

Missense mutations such as Glu270Lys, Tyr141Cys, and Gly511Arg were associated with endosomal impairments and enlarged endosomes; however, their exact disease-related mechanisms remain unclear. These mutations were not found to be associated with reduced *SORL1* expression [69]. The *SORL1* Arg953Cys mutation was identified in a familial AD case with TDP-43 pathology. The Arg953 residue was located near the conserved YWTD repeat sites of the beta-propeller domain in the SORL1 protein. Functional studies demonstrated that this mutation resulted in decreased *SORL1* expression on the cell surface, leading to impaired endosomal trafficking [70].

Studies with pluripotent stem cells (iPSCs) carrying a truncation mutation revealed reduced *SORL1* levels in the mutant cell lines. Reduced or lost *SORL1* levels were suggested to impair endosomal, lysosomal, and autophagy-related pathways [64,69,71]. Heterozygous *SORL1*-truncating mutations in human stem cell-derived neuronal models showed reduced lysosomal cathepsin D (CTSD) activity, abnormal autophagy, and impaired lysosomal clearance. Homozygous truncating mutations were found to result in more severe impairments in endo-lysosomal pathways and autophagy [65].

Mouse models were shown that *SORL1* heterozygous knockout mice presented an elevated degree of amyloid processing and amyloid deposition in the brain. Stimulating *SORL1* expression in mouse neurons and cell models led to the reorganization of endosomes and the Golgi network, as well as reduced amyloid processing. This study suggested that normal *SORL1* expression protects against amyloid processing and deposition in the brain [71].

To model the potential effects of truncating mutations, Göttingen minipig models were used, where heterozygous knockout of *SORL1* was generated by CRISPR-Cas9. Haploinsufficiency was confirmed in these pigs, as both *SORL1* mRNA and protein expression were reduced. Biomarker analysis revealed elevated levels of amyloid peptides (both short and long), and Tau proteins in the CSF during the preclinical phase of AD, which clearly reflected the biomarker changes observed in patients with preclinical AD. However, the ratio of Ab42/Ab40 did not change in the CSF of minipigs, suggesting that the amyloid deposition did not occur in them yet. No significant changes were detected via brain imaging of young adult minipigs. Pigs with truncating SORL1 mutations also exhibited enlarged endosomes, leading to impaired endosomal recycling activity. This study suggested that minipigs aged between 5 and 30 months are useful for modeling the preclinical and early phases of AD [71]. Figure 2 summarizes the effects outcomes from studies analyzing SORL1 haploinsufficiency in AD.

## 4. Haploinsufficiency of ABCA7

ABCA7 plays a role in lipid metabolism, cholesterol homeostasis, and phagocytosis-related pathways, which influence amyloid clearance. ABCA7 was shown to impact both early- and late-onset forms of AD through both gain-of-function and loss-of-function mechanisms. Studies on *ABCA7* knockout mice and cells revealed that *ABCA7* impacts APP processing and amyloid peptide production [72,73,74,75].

Nonsense, frameshift, or splice site mutations in *ABCA7* are thought to contribute to the disease through nonsense-mediated mRNA decay and haploinsufficiency. Although several missense mutations were identified in *ABCA7*, their roles and potential pathogenic mechanisms have yet to be fully investigated [76,77]. Mutations associated with premature termination codons (e.g., as Pro486fs, Gln1083Ter, Trp1214Ter, or Trp1336Ter) were relatively common in the Belgian cohort. Most of these mutations were associated with a positive family history, although segregation was not confirmed in all cases. This study did not provide any insight into whether these mutations impact *ABCA7* expression and related mechanisms [78]. Additionally, a large frameshift deletion (44 bp deletion, leading to a frameshift at Arg578) was suggested to be associated with AD, since it appeared more frequently in AD patients compared to the unaffected ones. This variant was more common in patients who had African ancestry. Transcriptome analysis revealed that the deletion resulted in detectable RNA strands for the mutant allele, suggesting a potential impact on protein function [79].

Allen et al. analyzed three loss-of-function variants in *ABCA7* (Leu1403fs, Gln709fs, c.5570+5G>C) and measured brain mRNA and protein levels using quantitative PCR and Western blotting. Carriers of the Leu1403fs variant presented with lower protein levels in their brains, but mRNA levels were not reduced. It was possible that the Leu1403fs mutation results in disease through loss-of-function mechanisms by inducing protein degradation or inefficient protein synthesis. The Gln709fs mutation reduced mRNA levels but not protein levels in the brain, suggesting that this mutation may not be associated with loss-of-function mechanisms, such as nonsense-mediated decay. However, the mutation was associated with putative gain-of-function mechanisms. Several patients were analyzed for the splice site mutation (c.5570+5G>C), but the data were inconsistent; some patients had low levels of ABCA7 protein in the brain, while mRNA levels remained unchanged. It was possible that other risk variants (e.g., common missense variants) may also influence *ABCA7* mRNA and protein levels in the brain [80].

Animal studies revealed that the heterozygous loss of *ABCA7* did not change amyloid cleavage or APOE levels but disturbed the amyloid uptake by macrophages and reduced amyloid plaque clearance [81]. Heterozygous knockout of *ABCA7* in mice led to reduced acute immune responses in the brain after stimulation with peripheral lipopolysaccharide (LPS). The expression of pro-inflammatory cytokines, such as IL-6, TNF, and IL-1β, was reduced in different brain areas, including the cortex and hippocampus. The inhibition of pro-inflammatory cytokines may be related to disturbances in the pathway during microglial activation. The *ABCA7* knockout resulted in reduced CD14 expression in the mouse brain and the accumulation of CD14 in microglial early endosomes. It was suggested that reduced ABCA7 levels disrupt the transport of CD14 in microglia during membrane trafficking. Also, *ABCA7* haploinsufficiency can inhibit the NF-κB pathway during the activation of microglia, leading to reduced pro-inflammatory cytokine production. Furthermore, *ABCA7* knockout mice were associated with amyloid-beta accumulation in microglia [82]. A recent study on zebrafish with *ABCA7* heterozygous knockout showed reduced immune functions, such as astroglial proliferation and microglial activation in response to amyloid peptide. *ABCA7* haploinsufficiency also decreased synaptic density. Gene expression studies revealed that reduced *ABCA7* expression resulted in a lower expression of the anti-inflammatory neuropeptide Y (NPY). Low NPY levels led to a decrease in the expression of the brain-derived neurotrophic factor (BDNF) and nerve growth factor receptor (NGFR), which are crucial for astrocyte, microglia, and synapse development. This study suggested that *ABCA7* plays a role in synaptic resilience through NPY expression [83,84]. Figure 3 summarizes the potential effects of *ABCA7* haploinsufficiency, based on animal and cell models.

## 5. Haploinsufficiency of *TREM2*

*TREM2* was identified as a risk factor for AD and other neurodegenerative diseases, including FTD and Nasu–Hakola disease (NHD). TREM2 protein was a receptor on the surface of microglia, and it regulates protein tyrosine phosphorylation outside the cells. Mutations in *TREM2* can lead to haploinsufficiency, abnormal microglial functions, and reduced amyloid clearance [85,86]. The *TREM2* Arg47His variant was studied in mouse models, which revealed the reduced expression of *TREM2*, as both mRNA and protein levels were lower. However, studies on induced pluripotent stem cells (iPSCs) derived from human AD patients with the *TREM2* Arg47His mutation failed to observe a reduced *TREM2* expression. The mutation was found to cause haploinsufficiency in mice by generating alternative splicing in *TREM2* through the addition of a cryptic splice site to the *TREM2* transcript. However, in humans, the splicing pattern of *TREM2* remained normal in the case of the Arg47His mutation [87].

In addition to missense variants, stop codon mutations have also been observed in patients with *TREM2* mutations linked to AD and other forms of dementia, potentially resulting in haploinsufficiency. The Gln33Ter mutation was reported in cases of NHD, AD, and FTD [88,89,90,91,92,93,94]. When transfected into HeLa cells, Western blot analysis showed no *TREM2* protein expression in cells with the Gln33Ter mutation [92]. Plasma analysis of patients with Gln33Ter revealed very low levels of soluble *TREM2* [90]. Another stop codon variant, Trp191Ter, was observed in African-American AD patients, but its pathogenic nature remains unclear. Patients with this mutation exhibited reduced levels of soluble TREM2. However, since this mutation exists only in the shortest *TREM2* transcript (transcript variant 2), its pathogenicity was still debated [93,94].

Multiple studies have investigated the potential effects of *TREM2* haploinsufficiency, but further research was needed on the AD-related mechanisms of having a single copy of *TREM2*, since conflicting results were reported. Ulrich et al. (2014) performed heterozygous *TREM2* knockout experiments in APPS1-21 mouse models. The amyloid deposition in *TREM2* haploinsufficient and normal *TREM2* mice was similar. However, plaque-associated microglial activation was significantly altered in the mice with one copy of *TREM2.* Immune markers, such as NOS2 and C1qa, were also reduced in *TREM2*+/− mice [95]. Wang et al. (2015) reported that *TREM2* haploinsufficiency and other dysfunctions increased amyloid deposition. Reduced *TREM2* expression and *TREM2* deficiency resulted in lower microglial activation and proliferation, leading to a higher degree of microglial loss [96]. Yuan et al. (2016) reported similar results in mice with *TREM2* haploinsufficiency; in their study, microglial activation was lower on amyloid plaques. Although the number of amyloid plaques did not change, the plaques became less compact, with larger surfaces. The larger plaque surface made them more accessible to neutrophils, leading to a higher degree of axonal dystrophy [97].

Delizannis et al. (2021) analyzed the *TREM2*+/+, *TREM2*+/−, and *TREM2*−/− microglial expressions in 5*FAD mouse models. Mice with *TREM2*+/− microglia were associated with reduced plaque-associated microglia, although the number was still higher than in the *TREM2*−/− mice. Gene expression in *TREM2*+/− 5*FAD mice showed a typical disease-associated microglial (DAM) gene expression profile in their brains, which was intermediate between those of the *TREM2*+/+ and *TREM2*−/− mice. Tau injection into the mice resulted in Tau pathology in both *TREM2*+/− and *TREM2*−/− mice. This study found that *TREM2*+/− mice were more effective in disease modeling compared to those with full *TREM2* knockout [98,99].

Sayed et al. (2018) showed that mice with *TREM2* haploinsufficiency experienced a higher degree of microglial activation-related injury compared to those with normal *TREM2.* In mouse models with Tau mutations, normal *TREM2* expression protected against Tau-associated microglial activation. However, in mice with *TREM2* haploinsufficiency, Tau pathology was exacerbated. It remained unclear whether microglial activation and Tau pathology interact in neurodegenerative processes or act independently [100]. *TREM2* haploinsufficiency was also suggested to impact the complement system. In mouse models expressing mutant Tau, heterozygous *TREM2* knockout resulted in increased complement-mediated microglial engulfment and accelerated synaptic loss. Administration of a 41 bp long *TREM2* peptide improved the condition of the mice by inhibiting complement activation [101].

However, the reduced *TREM2* expression was suggested to have minor protective effects as well. Schoch et al. (2021) inhibited *TREM2* expression in *APP/PSEN1* mice using antisense oligonucleotides (ASOs). Their findings showed a reduction in amyloid deposition during the early stages of AD, but the inhibition was ineffective in the later stages. These studies suggested that inhibiting *TREM2* in the early stages of the disease can slow down its progression, but not in later stages [102]. Figure 4 summarizes the potential effects of *TREM2* haploinsufficiency, based on animal and cell models.

## 6. Other AD Risk Genes with Possible Haploinsufficiency

As a complex disease, AD was extensively studied in various cell and animal models. Heterozygous knockout of different genes in AD mouse models was associated with either increased amyloid production or worsened amyloid pathology. Additionally, other mechanisms were reported, such as enhanced Tau phosphorylation and accumulation, altered microglial activation, abnormal autophagy, and reduced amyloid clearance (Figure 5) [103,104,105,106,107,108,109,110,111,112,113,114,115,116,117,118,119,120,121,122,123,124,125,126,127,128,129,130,131,132,133,134,135,136]. 

*ADAM10* plays a crucial role in the alpha-secretase cleavage of APP. A rare stop codon (Tyr167Ter) was identified in a familial case of AD, where family members presented with both early-onset and late-onset forms of the disease. Both full-length and truncated *ADAM10* isoforms were detected in the patient’s CSF, and normal *ADAM10* levels were found to be lower compared to the controls. Furthermore, levels of soluble APP were also reduced, suggesting that Tyr167Ter mutation can inhibit alpha-secretase cleavage, leading to increased amyloid peptide generation [103,104]. Mouse models revealed that two missense mutations in *ADAM10* (Gln170His and Arg181Gly) can also attenuate *ADAM10* activity in the human brain, resulting in elevated amyloid production and accumulation. These variants reduced ADAM10’s chaperone activity, leading to lower neuroprotection. Additionally, decreased *ADAM10* expression was observed in the brains of mutant mice, suggesting that haploinsufficiency cannot be ruled out for these variants [104,105].

The angiotensin-converting enzyme (*ACE*) was an enzyme involved in the conversion of angiotensin I to angiotensin II, and it impacts the cleavage of long amyloid peptides to shorter amyloid peptides in the brain. *ACE* was shown to be protective against AD by reducing amyloid aggregation. Inhibition of *ACE* and *ACE* variants was increased the risk of AD onset. *ACE* was also suggested to protect against AD by mitigating metal-induced oxidative damage [106,107]. Xie et al. (2021) analyzed the role of frameshift and stop-gain *ACE* mutations in late-onset AD patients through the Shanghai FLOAD study. In vitro studies revealed that cell cultures with Leu1042fs or Leu1042Ter mutations were associated with the reduced expression of membrane-localized and secreted ACE proteins. However, the mutations did not affect the processing of full-length APP proteins [108]. Heterozygous *ACE* deletion or *ACE* inhibition in AD (hAPPSw) mice resulted in an elevated degree of amyloid deposition in senile plaques and an increased Aβ42/40 ratio. Additionally, *ACE* haploinsufficiency accelerated the loss of nerve cells [107]. Danilov et al. (2024) studied potential loss-of-function *ACE* mutations (e.g., Arg120fs, Trp314Ter, Asp412fs, Ser1209Profs) in AD patients, which were associated with reduced levels of ACE in the blood. Furthermore, missense mutations, such as Tyr215Cys, lowered ACE protein levels in the patient’s blood. Heterozygous stop-gain or frameshift mutations resulted in truncation, reduced *ACE* expression and activity, and potentially were associated with reduced Aβ42 hydrolysis. Missense mutations in the N-domain of the active site in the ACE protein also impaired amyloid hydrolysis. Alternatively, mutations, especially C-terminal mutations, decreased ACE transport and surface expression, leading to lower enzyme function [109].

The vacuolar protein-sorting ortholog 35 (*VPS35*) was involved in retromer endosomal trafficking, impacting transport from endosomes to the trans-Golgi network or from endosomes to the plasma membrane. The role of *VPS35* was verified in multiple neurodegenerative diseases, including AD, ALS, and Parkinson’s disease (PD) [110]. Mouse models suggest that *VPS35* may be involved in AD pathogenesis through haploinsufficiency. Hemizygous *VPS35* deletion in Tg2576 mice was associated with earlier disease onset and cognitive dysfunction. Long-term potentiation and postsynaptic glutamatergic neurotransmission were impaired in *VPS35*-deficient mice, while amyloid deposits in their brains were increased. *VPS35* haploinsufficiency also accelerated beta-secretase activity and enriched BACE1 in endosomes [111]. Furthermore, *VPS35* knockout induced abnormal microglial functions, such as disease-associated microglia development and impaired amyloid clearance [112].

Mitochondrial deacetylase sirtuin-3 (*SIRT3*) was an AD risk factor that functions as a NAD-dependent histone deacetylase localized in the inner membrane and matrix of mitochondria. *SIRT3* was involved in multiple pathways, including the regulation of ROS-related pathways, fat metabolism, mitochondrial antioxidants, and increased neural survival. *SIRT3* expression was reduced in the cerebral cortex of AD patients [113]. *SIRT3* haploinsufficient *APP/PSEN1* mice showed a loss of gamma-aminobutyric acid (GABA)-ergic parvalbumin and calretinin interneurons in the cortex. Furthermore, *SIRT3* knockout resulted in epileptic seizures in mice, potentially by exacerbating neuronal loss and oxidative stress. A diet rich in ketone ester improved the condition of the mice by restoring *SIRT3* expression [114,115].

Shugoshin-1 (*SGO1*) was a cell cycle-regulator protein that plays a significant role in chromosomal segregation during cell division. While *SGO1* dysfunctions were suggested to impact cancers [116], their role in AD should not be overlooked either [117]. Mouse models with heterozygous knockout of the *SGO1* gene presented typical AD pathology, including amyloid peptide and Tau accumulation. Heterozygous *SGO1* knockout mice showed reduced levels of *SGO1* expression, leading to prolonged mitosis due to chromosome-cohesion defects and abnormal centrosome integrity. It was possible that *SGO1* dysfunctions interact with other AD-causative or risk variants, contributing to disease pathology. However, the direct role of *SGO1* haploinsufficiency in cognitive dysfunctions at later disease stages should not be ruled out [118].

*PICALM* was identified as an AD risk factor and functions as a clathrin-adaptor protein. *PICALM* impacts clathrin-mediated endocytosis and autophagy, and abnormal *PICALM* affects both amyloid processing and Tau pathology [119,120,121]. Reduced *PICALM* expression in the brain endothelium of AD patients was associated with cognitive dysfunctions, amyloid deposits, and AD neuropathology. Inducing *PICALM* expression increased amyloid clearance in AD mouse models [120,121]. *PICALM* haploinsufficiency was also suggested to be linked to Tau dysfunctions. *PICALM* knockout mice (Tg30xPicalm+/−) exhibited a high degree of age-related Tau deposits and neurofibrillary tangles in their brains, along with increased autophagy markers [122].

Activity-dependent neuroprotective protein (*ADNP*) interacts with multiple proteins involved in chromatin remodeling, RNA splicing, microtubule dynamics, and autophagy. Overall, *ADNP* plays a role in neuroprotection. *ADNP* dysfunctions are linked to multiple neurological diseases, including autism, schizophrenia, PD, and AD [123]. ADNP-haploinsufficient mouse models exhibited cognitive dysfunctions and Tau-related pathology. Reduced *ADNP* expression was suggested to impair microtubule dynamics, including Tau dynamics [124].

Myeloid Arginase 1 (*ARG1*) was involved in arginine hydrolysis, playing a role in the urea cycle and regulating immune mechanisms [125]. In AD mouse models, *ARG1* contributed to amyloid peptide clearance during IL-1β-related inflammatory pathways [126]. *ARG1* haploinsufficiency in myeloid cells of AD mice was associated with elevated amyloid deposition, impaired microglial activation (reduced phagocytosis), and altered behavior (anxiety, reduced fear, or fear-associated memory). The exact mechanism of *ARG1* haploinsufficiency and its impact on phagocytosis remains unclear, but it was suggested that reduced *ARG1* expression can disturb mTORC1 inhibition-related autophagy [127].

The JNK-interacting protein 3 (*JIP3*) was involved in the regulation of axonal lysosome maturation, transport, and abundance. Knocking out *JIP3* in heterozygous AD mouse models worsened amyloid plaque pathology, suggesting that lower *JIP3* expression escalates the levels of soluble amyloid peptides, leading to larger plaques and increased axonal dystrophy. *JIP3* knockout mice presented an accumulation of lysosomes in axons, similar to amyloid plaques. This study revealed that *JIP3* haploinsufficiency resulted in impaired APP processing, reduced amyloid clearance, and intraneuronal amyloid peptide generation [128].

*X11alpha* was a cytoplasmic adaptor protein. Stimulating the expression of X11 proteins was protective against APP processing and amyloid production. However, conflicting results were reported regarding the impact of X11alpha and X11beta inhibition on APP metabolism [129,130]. Heterozygous knockout of *X11alpha* in AD mouse models resulted in elevated levels of APP C-terminal fragments and soluble amyloid peptides in the brains of these mice. These findings suggest that *X11alpha* slows down APP metabolism through endo-lysosomal trafficking. Reduced *X11alpha* expression increased APP trafficking, resulting in higher amyloid peptide generation by beta and gamma secretases [131].

Mesenchyme homeobox 2 (*MEOX2*) was involved in vascular differentiation, but its role in AD remains unclear. *MEOX2* expression was reduced in the endothelial brain cells of AD patients. *MEOX2*-haploinsufficient mice exhibited similar amyloid plaque deposits compared to AD mouse models with normal *MEOX2* expression. However, the degree of neuronal loss was higher in mice with reduced *MEOX2* expression, especially near plaque areas, suggesting that *MEOX2* haploinsufficiency decreased the degree of neuroprotection. *MEOX2* haploinsufficiency was also associated with reduced vascular density, which may enhance neuronal loss and impact AD pathology [132].

*CTSD* was an aspartyl proteinase that was identified as a risk factor for several neurodegenerative diseases, including AD, PD, and dementia with Lewy bodies. Cathepsins, including *CTSD*, play a significant role in neuronal homeostasis, controlling protein processing and lysosomal degradation. Loss-of-function mutations and the reduced expression of *CTSD* were associated with elevated amyloid deposition. *CTSD* was involved in the degradation of amyloid peptides and can regulate the Aβ42/40 ratio. The inhibition of *CTSD* in mice was associated with increased levels of amyloid peptides. AD mouse models with *CTSD* haploinsufficiency also exhibited elevated amyloid deposition. Furthermore, *CTSD* knockout in mice was associated with tauopathy, suggesting that *CTSD* also impacts abnormal Tau degradation [133,134,135].

Telomerase reverse transcriptase (*TERT*) was involved in maintaining telomerase-related functions and controlling stress responses and chromatin organization. *TERT* deficits were suggested to contribute to neurodegenerative phenotypes. Mouse models (3xTg-AD, 5xFAD) with *TERT* haploinsufficiency presented increased *APP* expression and reduced *BDNF* expression, leading to elevated amyloid deposition and cognitive decline. *TERT* induction reduced amyloid peptide levels in neurons of AD mouse models and improved cognition via the beta-catenin/TCF7 complex [136].

## 7. Haploinsufficiency and Neuroprotection in AD

Besides the involvement in neurodegenerative processes, heterozygous knockout of certain genes, such as *APOE*, *BACE1*, or Inositol polyphosphate-5-phosphatase (*INPP5D*), was found to be protective against AD by slowing down amyloid progression, reducing amyloid deposition, and diminishing immune-related mechanisms (Figure 6) [137,138,139,140,141,142,143,144,145,146,147,148,149,150,151,152,153,154].

The *APOE* E4 allele was a well-established risk factor for late-onset AD. It was suggested that genetic modifications of the *APOE* E4 allele can protect against amyloid deposition and AD progression. Loss-of-function mutations in *APOE were* reported to be rare worldwide. However, premature stop codon mutations in *APOE* were analyzed using data from the Alzheimer’s Disease Sequencing Project (ADSP), revealing truncation variants, including Trp5Ter, Leu8Ter, and Gln39Ter, in cognitively healthy individuals aged 71 to 90 years. Most of these individuals had the *APOE* E3 allele as their effective *APOE*. One patient with the *APOE* Trp5Ter and *APOE* E3/E4 genotype exhibited resistance to amyloid pathology but not to Tau pathology. However, another cognitively normal carrier of the *APOE* Trp5Ter mutation had elevated levels of amyloid peptides in the CSF. Interestingly, a patient with late-onset AD (75 years old) carried a deletion (19:44905303–44907102) that included the promoter and exons 1–2 of the *APOE* gene. This patient’s effective *APOE* was *APOE* E4, and their age of onset was similar to those with the *APOE* E3/E4 genotype [137,138,139]. Mouse models with heterozygous *APOE* knockout (*APOE* E3/− and *APOE* E4/−) presented reduced *APOE* expression, leading to decreased amyloid processing and deposition, as well as lower microglial activation [140]. Another study showed that heterozygous *APOE* E3 and E4 knockout did not affect amyloid clearance in six-month-old mice. However, at twelve months, mice with reduced *APOE* dosage had fewer plaques and lower amyloid peptide levels [141]. Additionally, reducing the *APOE* E4 dosage was suggested to protect against Tau pathology, reduce neuroinflammation, and lower NfL levels in the plasma [140]. Knocking out *APOE* protected against AD pathology without affecting cholesterol metabolism, suggesting that *APOE* silencing could be a useful approach to slow or delay disease progression. However, further studies are needed to understand how *APOE* modulation impacts neurodegenerative pathways [142].

*BACE1* was involved in APP cleavage and impacts amyloid peptide production. However, conflicting results exist regarding whether *BACE1* haploinsufficiency reduces amyloid processing. Devi and Ohno (2013) observed similar amyloid plaque deposition and memory dysfunction in *BACE1*-haploinsufficient mice compared to normal mice, suggesting that *BACE1* haploinsufficiency does not attenuate amyloid processing and deposition in familial AD (FAD) mouse models. *BACE1* haploinsufficiency did not prevent the expression of amyloid-reducing enzymes like neprilysin [143]. In another study, *BACE1* haploinsufficiency resulted in lower levels of amyloid oligomers, Aβ42 peptides, and plaque burden in 6–7-month-old 5*FAD mouse models (both male and female). These mice also had higher levels of APP and soluble APP-alpha. However, at 15–18 months, the amyloid oligomers and C99 levels were elevated in mice with *BACE1* haploinsufficiency. *BACE1* haploinsufficiency was suggested to slow down memory dysfunction in AD mouse models [144].

*GRN* regulates several processes, including neuronal growth, microglial functions, inflammation, tumorigenesis, and wound healing. *GRN* variants, particularly frameshift and stop-gain mutations, were identified as causative factors for FTD. However, *GRN* variants (e.g., rs5848) have also been suggested to be risk factors for AD. Studies on APP-transgenic mice showed that heterozygous *GRN* knockout reduced amyloid deposition. However, lower *GRN* expression induced pathogenic mechanisms independent of amyloid, such as increased Tau deposition through CDK activation or lysosomal dysfunction [145].

*INPP5D* was a lipid phosphatase and signal transduction regulator in the immune system. It was identified as an inhibitor of TREM2 and a regulator of microglial functions. Variants in *INPP5D* were suggested as AD risk factors. Heterozygous *INPP5D* knockout in 5xFAD mice revealed beneficial effects on amyloid deposition, with *INPP5D* haploinsufficiency protecting cognitive function in these models. Additionally, reduced *INPP5D* expression enhanced microglial functions and phagocytosis by activating TREM2 and increasing amyloid peptide clearance [146].

Triggering Receptor Expressed on Myeloid Cells-1 (*TREM1*) was an inducer of pro-inflammatory molecules and reactive oxygen species (ROS) in response to damage or infection. TREM1 deficiency renders microglia resistant to amyloid oligomers, leading to reduced amyloid clearance. The 5xFAD mouse models with heterozygous *TREM1* knockout demonstrated reduced memory dysfunction and preserved microglial homeostasis. Furthermore, experiments on Swedish *APP* mice showed that *TREM1* haploinsufficiency prevents memory dysfunction by protecting against mitochondrial damage and abnormal glucose metabolism in the brain [147].

Transmembrane protein 59 (*TMEM59*) was a Type-I glycoprotein expressed in multiple tissues. *TMEM59* overexpression was suggested to induce amyloid processing by retaining APP in the Golgi. Mice with *TMEM59* overexpression exhibited memory dysfunction similar to that observed in wild-type mice and demonstrated elevated amyloid levels. *TMEM59* ubiquitously haploinsufficiency in familial AD (FAD) mouse models resulted in reduced amyloid plaques and amyloid peptide levels, as well as improved synaptic plasticity and memory. These findings suggest that *TMEM59* inhibition could be a potential therapeutic strategy for AD [148].

Insulin-like growth factor-1 (*IGF-1*) modulation was suggested to play a role in neuroprotection and the prevention of AD. Both insulin and *IGF*-1 impact amyloid trafficking in the blood and CSF. Mouse models with *IGF-1* haploinsufficiency displayed delayed neurodegeneration and slowed amyloid accumulation. Further studies are needed to investigate the role of *IGF-1* and insulin receptor modulation in neuroprotection [149].

Glutamic acid decarboxylase 67 (*GAD67*) was responsible for converting glutamate into GABA. While *GAD67* may impact various neurological diseases, such as schizophrenia and epilepsy, its role in AD was not thoroughly investigated. AD mouse models with *GAD67* haploinsufficiency showed significant reductions in amyloid production, abnormal GABA accumulation, abnormal tonic inhibition, and microglial activation. Furthermore, *GAD67* haploinsufficiency protected against olfactory impairment in AD mice, suggesting that *GAD67* may be a potential therapeutic target candidate for AD [150].

Synaptojanin 1 (*SYNJ1*) was a phosphoinositol (4,5)-biphosphate phosphatase enzyme implicated in the onset of AD through the regulation of endocytic traffic within synapses. Cell models (N2a) with *SYNJ1* inhibition showed reduced amyloid peptide generation. Additionally, *SYNJ1*-haploinsufficient AD mouse models (with Swedish *APP* and *PSEN1* exon 9 deletion) exhibited a lower plaque load, reduced amyloid peptide levels, and improved cognition. Lower *SYNJ1* expression did not affect APP processing by beta- and gamma-secretases but increased amyloid clearance through enhanced cellular uptake [151].

Hexokinase 2 (*HK2*) plays a role in microglial metabolism and the control of inflammatory pathways. *HK2* expression was increased in the brains of AD patients. *HK2* haploinsufficiency in AD mice was associated with reduced inflammatory pathways. Heterozygous *HK2* knockout slowed down disease progression through the IKBα-NF-κB microglial pathway. However, complete deletion of *HK2* did not improve the condition of the mice; furthermore, it caused additional damage, such as mitochondrial deficits [152,153].

Protein kinase RNA-like ER kinase (*PERK*) was a eukaryotic initiation factor-2alpha (eIF2α) kinase involved in protein folding. Abnormal phosphorylation and elevated eIF2α levels can induce AD through BACE1 and cAMP-response element-binding protein (CREB) expression. Heterozygous *PERK* knockout in AD mouse models inhibited overactivation of the ERK-eIF2α pathway, reducing memory damage in the mice. Additionally, it prevented amyloid production by decreasing *BACE1* expression and restoring *CREB* function [154].

## 8. Discussion

The aim of this review article was to discuss the potential role of haploinsufficiency in AD. AD was a complex disease in which several genetic and environmental factors were identified as contributing to disease onset [12]. Mutations involved in AD can be associated with both gain-of-function and loss-of-function mechanisms [12]. Studies from animal experiments indicate that the role of haploinsufficiency in AD should not be overlooked. Haploinsufficiency was shown to contribute to the onset of different neurodegenerative diseases. For instance, repeat expansions in the promoter of the *c9orf72* gene are related to both gain-of-function and loss-of-function mechanisms, leading to ALS or FTD. The haploinsufficiency of c9orf72 can cause glutamate receptor accumulation and excitotoxicity in response to glutamate, ultimately resulting in the reduced survival of motor neurons. Furthermore, reduced c9orf72 expression leads to increased sensitivity of nerve cells to dipeptide repeat protein (DRP) toxicity and diminished DRP clearance [11,155]. Additionally, *GRN* haploinsufficiency induces microglial dysfunction, TDP43 aggregation, and impaired lysosomal function, contributing to FTD [156]. Besides their neurodegenerative effects, haploinsufficiency of certain genes were found to be protective against neurodegeneration. The heterozygous knockout or inhibition of sterile alpha and TIR motif-containing 1 (*SARM1*), a gene encoding a pro-degenerative NADase, was shown to slow down axonal death [157]. Moreover, the haploinsufficiency of Ran-binding protein-2 (*RanBP2*) was reported to prevent apoptosis and membrane dysgenesis under light-related stress in the eye [158].

Studying haploinsufficiency in complex diseases, including AD, should be important. Identifying mutations associated with reduced gene expression can provide deeper insights into neurodegenerative mechanisms. Additionally, analyzing haploinsufficiency may yield benefits for therapeutic interventions. Multiple methods were developed to measure the impact of mutations on mRNA and protein levels, including reverse transcription PCR, quantitative PCR (RT-qPCR), and RNA sequencing (RNAseq) for quantifying mRNA from blood or brain tissue samples from patients [159,160]. Haploinsufficiency can be detected by quantifying protein levels through Western blotting, ELISA, or mass spectrometry [161]. It can also be modeled in cellular or animal models (e.g., 5xFAD mice in AD). Techniques such as CRISPR-Cas9, antisense oligonucleotides (ASO), and RNA interference are common methods to mimic the effects of heterozygous gene knockout [2,21,80,162,163]. In the context of AD, certain genes, such as *ABCA7* [80] and *SORL1* [71], were confirmed to impact neurodegeneration through haploinsufficiency. Animal studies have indicated that heterozygous deletions of genetic factors, such as *PICALM* [119], *VPS35* [110,111,112], or *CTSD* [133,134,135] can induce loss-of-function mechanisms, resulting in decreased neuroprotection, reduced amyloid clearance, or enhanced microglial activation. Interestingly, loss-of-function *PSEN1* mutations (including *PSEN1* Ala246Glu and *PSEN1* Cys410Tyr) were associated with an elevated long amyloid/short amyloid ratio due to the partial inhibition of short amyloid production. Moreover, *PSEN* haploinsufficiency was resulted in neurodegeneration through non-amyloid-related mechanisms, including reduced synaptic plasticity, altered Notch signaling, and Tau phosphorylation [32,33,35]. Restoring appropriate expression levels of these genes was shown to enhance amyloid clearance and neuroprotection [91,96]. Conversely, studies on AD mouse models involving the heterozygous knockout of genes, such as *APOE* [137,138,139,140,141,142], *PERK* [154], or *TMEM59* [148] were associated with neuroprotective mechanisms [115,117]. Heterozygous knockout of other genes, including *BACE1* [143], *TMEM59* [148], *GAD67* [150], or *SYNJ1* [151], was also associated with a lower degree of amyloid deposition, increased neuron survival, or accelerated amyloid clearance. Taken together, understanding how haploinsufficiency could affect the neurodegeneration or neuroprotection should be important to find out the disease-related mechanisms and in the identification of disease-related biomarkers.

Furthermore, understanding the role of haploinsufficiency in AD-causing risk genes or potential candidates can open new avenues in drug discovery for AD. Finding out whether these genes may impact AD through haploinsufficiency could provide possible new therapeutic targets for the disease. Identifying patients who carry haploinsufficiency-associated variants can enhance the development of personalized medicines. CRISPR-Cas9 and other gene-editing techniques may offer useful approaches for correcting mutations involved in AD [164]. ASOs are also promising for developing therapeutic strategies for AD and other neurodegenerative diseases. By modulating gene expression at the RNA level, they can induce the expression of the wild-type allele and restore the production of functional proteins [165,166].

However, current studies on haploinsufficiency in diseases have several limitations. First, the majority of studies were performed on model organisms (e.g., mice) or cell lines, which may not accurately reflect the effects of haploinsufficiency in humans. There may be differences in the gene expression patterns of the same mutations between animal models and humans. For example, the *TREM2* Arg47His mutation resulted in reduced *TREM2* expression in mice but not in humans [85,166,167,168]. Another issue was that haploinsufficiency of certain genes can result in diverse phenotypes. Besides AD, truncation mutations in *PSEN1* and *PSEN2* were associated with a wide range of disease phenotypes, such as FTD, acne inversa, and acute encephalopathy. This wide variety of phenotypes can complicate disease modeling [40,41,42,43,44,45,46,47,48,49,50,51,52,53]. Studying haploinsufficiency in humans also presents significant challenges due to the heterogeneity of the human population. Individuals from different populations often have diverse genetic backgrounds that can influence disease phenotypes [169]. Modifier genes and environmental factors may also affect the expression of haploinsufficient genes. In animal models, it can be difficult to model the impact of these genetic modifiers and gene–environment interactions concerning haploinsufficiency [170,171,172].

There are also technical challenges associated with studying haploinsufficiency. Detecting gene dosage can be difficult, and sensitive techniques (e.g., quantitative PCR or RNA sequencing) may be expensive and not accessible to everyone [173]. Furthermore, measuring the levels of certain proteomic markers, including membrane proteins (such as *PSEN*s, or *ABCA7*) may be challenging. Membrane proteins are less abundant in biological fluids and difficult to measure by conventional proteomic technologies. It may be challenging to isolate membrane proteins since they can lose their stability. However, these membrane proteins can be cleaved by proteolytic enzymes. The fragment of membrane proteins released to the plasma can serve as alternative disease biomarkers [174,175,176]. Although CRISPR-Cas9 was a promising technique for modeling diseases, its results can be misleading due to the potential for off-target effects [177]. In the future, advanced gene-editing technologies may open new avenues for haploinsufficiency research in various diseases, including AD. Increased specificity and reduced off-target effects should enhance the ability to study the potential impacts of haploinsufficiency [177,178,179]. Also, the utilization of biomarkers for haploinsufficiency should be essential for the disease diagnosis and in understanding the possible disease-associated pathways [180]. Multi-omics approaches (including genomics, transcriptomics, proteomics, and metabolomics) may also be useful for discovering how altered dosage influences AD-related pathways [181]. The development of humanized animal models could yield more accurate insights into disease modeling [182].

In conclusion, reduced gene dosage can play a significant role in AD onset and even in protection against the disease. Therefore, studying haploinsufficiency in these models can provide valuable insights into disease-related pathways and contribute to the discovery of new therapeutic strategies for AD [58,59,60,61,62,63,64,65,66,67,68,69,70,71,72,73,74,75,76,77,78,79,80,81,82,83,84,85,86,87,88,89,90,91,92,93,94,95,96,97,98,99,100,101,102,103,104,105,106,107,108,109,110,111,112,113,114,115,116,117,118,119,120,121,122,123,124,125,126,127,128,129,130,131,132,133,134,135,136,137,138,139,140,141,142,143,144,145,146,147,148,149,150,151,152,153,154]. Targeting haploinsufficiency through gene therapies or protein replacement, such as inducing gene expression with ASOs or CRISPR-Cas9, shows promise in treating patients with neurodegenerative diseases [183,184,185]. Gene therapies have already been successfully implemented for spinal muscular atrophy (*SMA1*), and therapies for other neurodegenerative diseases, including AD, PD, ALS, FTD, and Huntington’s disease (HD), are currently under development [185,186].

## Figures and Tables

**Figure 1 ijms-25-11959-f001:**
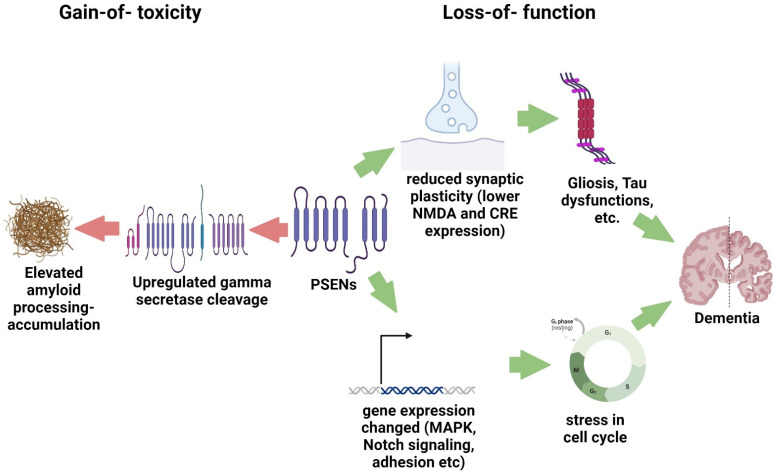
Gain-of-function vs. loss-of-function mechanisms of *PSEN* dysfunctions, leading to neurodegeneration.

**Figure 2 ijms-25-11959-f002:**
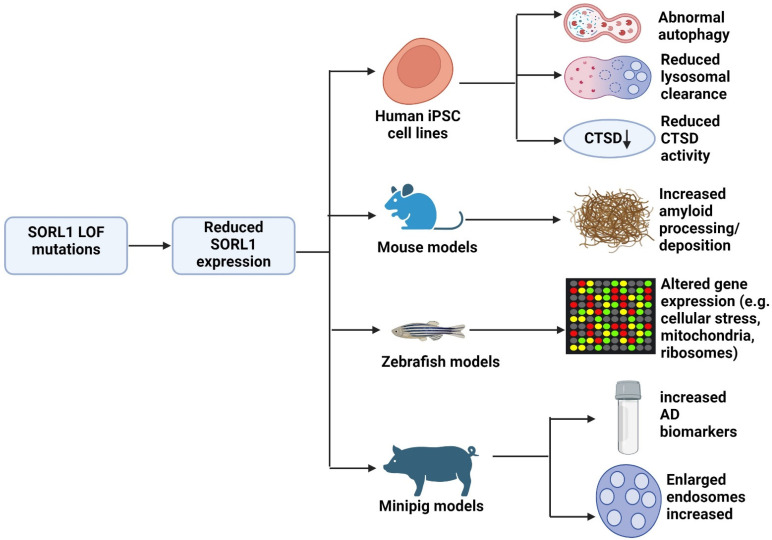
Potential effects of SORL1 haploinsufficiency, based on cell and animal models.

**Figure 3 ijms-25-11959-f003:**
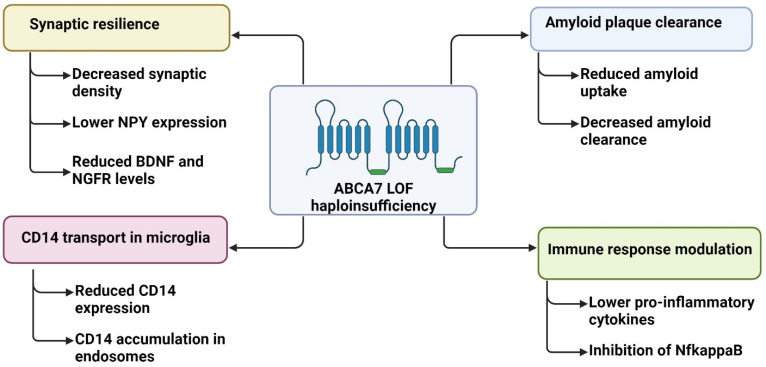
Possible effects of ABCA7 haploinsufficiency, based on experiments on cell and animal models.

**Figure 4 ijms-25-11959-f004:**
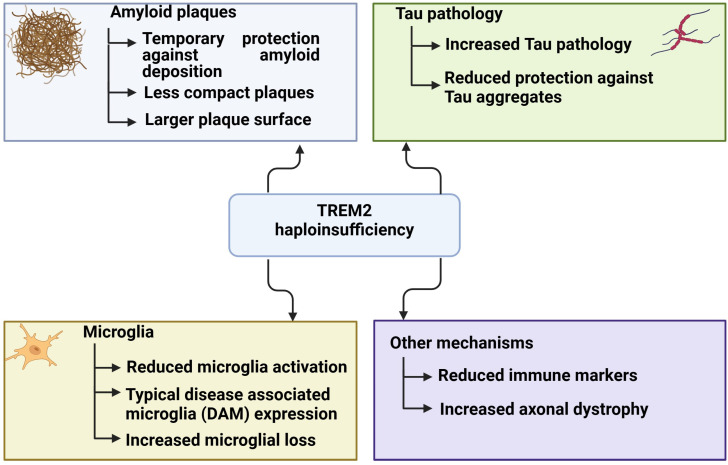
Potential effects of *TREM2* haploinsufficiency, based on animal and cell models.

**Figure 5 ijms-25-11959-f005:**
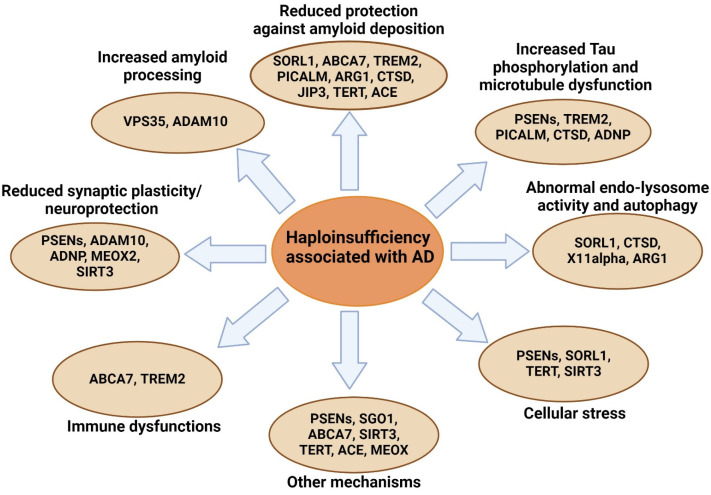
Possible mechanisms associated with AD haploinsufficiency.

**Figure 6 ijms-25-11959-f006:**
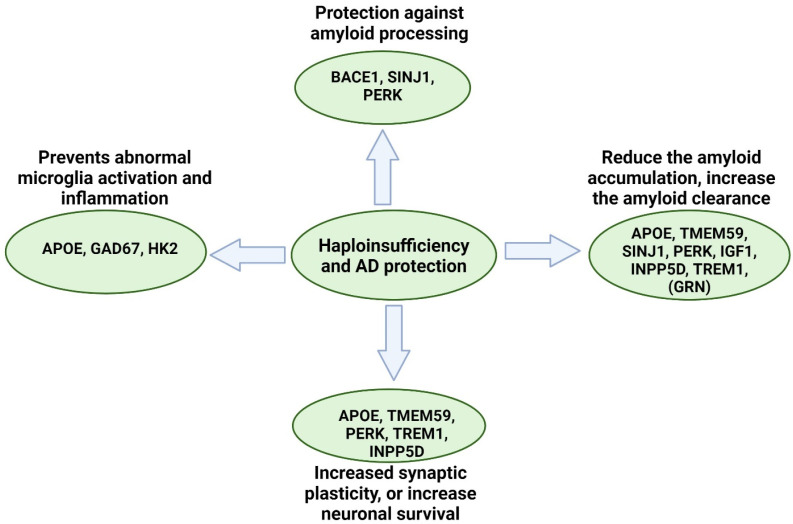
Potential beneficial effects of haploinsufficiency in AD.

**Table 2 ijms-25-11959-t002:** Examples of *SORL1* mutations, which were proven to result in AD through haploinsufficiency.

Mutation	Disease	AOO	Family History	Amyloid Changes	Transcript/Protein Changes	References
Gly447Argfs*22	AD	64	Probable positive	NA	Reduced SORL1 mRNA in patient’s lymphoblast	[62]
Arg744Ter	AD	NA	NA	Defects in Ab uptake by microglia	SORL1 expression was abolished	[63]
His962Profs*45	AD	52	NA	NA	Reduced SORL1 expression in iPSC cells, enlarged endosomes	[64]
Cys1431fs*2	AD	60s	Probable positive	Elevated APP accumulation in endosomes	Reduced SORL1 expression	[65,66]
Cys1478Ter	AD	60s	Positive but not segregating	NA	Reduced SORL1 expression, increased cellular stress	[66,67]
Trp1821Ter	AD	NA	NA	Positive	Reduced SORL1 expression, changes in mitochondrial and ribosomal functions	[68]

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
