# Peer review of "Haploinsufficiency and Alzheimer’s Disease: The Possible Pathogenic and Protective Genetic Factors"

_ijms, 2024, doi:10.3390/ijms252211959_

Round 1
Reviewer 1 Report
Comments and Suggestions for Authors
The current article discusses that how haploinsufficiency could contribute to Alzheimer’s disease pathogenesis. This is an interesting area, the article has few short comings that should be addressed before it can be accepted for publication.
1. Interestingly, heterozygous knockout of 86 certain genes, such as APOE, has protective effects against AD progression. Check for consistent usage of AD throughout the manuscript. The authors must see about the erratic abbreviation usage. At first places, all abbreviations must be mentioned.
2. Introduction section must be updated with latest information about Alzheimer’s disease 10.1016/j.molliq.2021.116888;
3. Figure 2. Potential effects of SORL1 haploinsufficiency, based on cell and animal models. Altered gene expression, make sure the figure used is original.
4. Figure 4. Potential effects of TREM2 haploinsufficiency, based on animal and cell models. I would suggest to represent in a better way instead of mentioning the author names.
5. Limitations and future perspectives of this article needs to be added. The authors must add a paragraph about the limitation and future perspective of the current research.
6. Table 2. Examples on SORL1 mutations, which were proven to result in AD through haploinsufficiency. Check all the references if they are correctly cited or not.
7. Figure 3. Possible effects of ABCA7 haploinsufficiency, based on experiments on cell- and animal models. NF-κB must be mentioned.
Author Response
The current article discusses that how haploinsufficiency could contribute to Alzheimer’s disease pathogenesis. This is an interesting area, the article has few short comings that should be addressed before it can be accepted for publication.
Thank you for the positive and constructive comments, we tried to revise your manuscript, according to your suggestions.
1. Interestingly, heterozygous knockout of certain genes, such as APOE, has protective effects against AD progression. Check for consistent usage of AD throughout the manuscript. The authors must see about the erratic abbreviation usage. At first places, all abbreviations must be mentioned.
Thank you, we fixed this issue, and added the abbreviation to the side, where they were mentioned first.
2. Introduction section must be updated with latest information about Alzheimer’s disease
Thank you. We added a paragraph with recent information about Alzheimer’s disease.
“Besides the pathogenic or risk variants, several neuroprotective factors were discovered against AD. As mentioned earlier, the most well-known protective variant is the APOE E2 allele. However, additional protective variants were discovered, more notably, the Christchurch variant in APOE (Arg136Ser) was found to prevent or slow down the Tau pathology, inflammation, or neurodegeneration [18, 19]. Additionally, an APP variant, Ala673Thr was found to protect against AD-related neurodegeneration and dis-ease-related alterations of biomarkers in cerebrospinal fluid (CSF) or blood [20]. Further-more, variants in other genes, including ABCA7 (Gly215Ser, Val1613Met) or fibronectin-1 (FN1) were found to protective against AD progression [21, 22, 23].
Several attempts were performed to develop drugs against Alzheimer’s disease. Majority of the available drugs (including acetyl -choline inhibitors) are focusing on reducing the disease symptoms, however, they cannot prevent the disease progression. Two drug candidates, aducanumab and lecanemab were shown disease-modifying effects in clinical trials, however, the further studies are needed to verify their long-term effects and safety [24, 25, 26]. Development of gene therapy against neurodegenerative diseases, including AD may also be promising for patients [27, 28].
3. Figure 2. Potential effects of SORL1 haploinsufficiency, based on cell and animal models. Altered gene expression, make sure the figure used is original.
Thank you for your comment. We created the entire figure ourselves using BioRender. The gene expression figure is a grouped icon included within the software.
4. Figure 4. Potential effects of TREM2 haploinsufficiency, based on animal and cell models. I would suggest to represent in a better way instead of mentioning the author names.
Thank you, we made a new figure on TREM2 haploinsufficiency.
5. Limitations and future perspectives of this article needs to be added. The authors must add a paragraph about the limitation and future perspective of the current research.
Thank you for the suggestion. A paragraph was added in the Discussion chapter.
“However, current studies on haploinsufficiency in diseases have several limitations. First, the majority of studies have been performed on model organisms (e.g., mice) or cell lines, which may not accurately reflect the effects of haploinsufficiency in humans. There may be differences in the gene expression patterns of the same mutations between animal models and humans. For example, the TREM2 Arg47His mutation resulted in reduced TREM2 expression in mice but not in humans [85, 166, 167, 168]. Another issue is that haploinsufficiency of certain genes can result in diverse phenotypes. Besides Alzheimer's disease (AD), truncation mutations in PSEN1 and PSEN2 have been associated with a wide range of disease phenotypes, such as frontotemporal dementia (FTD), acne inversa, and acute encephalopathy. This wide variety of phenotypes can complicate disease modeling [40-53]. Studying haploinsufficiency in humans also presents significant challenges due to the heterogeneity of the human population. Individuals from different populations often have diverse genetic backgrounds that can influence disease phenotypes [169]. Modifier genes and environmental factors may also affect the expression of haploinsufficient genes. In animal models, it can be difficult to model the impact of these genetic modifiers and gene-environment interactions concerning haploinsufficiency [170, 171, 172].
There are also technical challenges associated with studying haploinsufficiency. Detecting gene dosage can be difficult, and sensitive techniques (e.g., quantitative PCR or RNA sequencing) may be expensive and not accessible to everyone [173]. Furthermore, measuring the levels of certain proteomic markers, including membrane proteins (such as APP, PSENs or ABCA7) may be challenging. Membrane proteins are less abundant in bio-logical fluids, and difficult to be measured by conventional proteomic technologies. It may be challenging to isolate membrane proteins, since they can lose their stability. However, these membrane proteins can be cleaved by proteolytic enzymes. The fragment of membrane proteins, released to the plasma can be serve as alternative disease biomarkers [174, 175, 176]. Although CRISPR-Cas9 is a promising technique for modeling diseases, its results can be misleading due to the potential for off-target effects [177]. In the future, advanced gene-editing technologies may open new avenues for haploinsufficiency research in various diseases, including AD. Increased specificity and reduced off-target effects should enhance the ability to study the potential impacts of haploinsufficiency [177, 178, 179]. Also, the utilization of biomarkers for haploinsufficiency should be essential for disease diagnosis and in understanding the possible disease-associated pathways [180]. Multi-omics approaches (including genomics, transcriptomics, proteomics, and metabolomics) may also be useful for discovering how altered dosage influences AD-related pathways [181]. The development of humanized animal models could yield more accurate insights into disease modeling [182].’
6. Table 2. Examples on SORL1 mutations, which were proven to result in AD through haploinsufficiency. Check all the references if they are correctly cited or not.
We re-organized the reference. We will fix it.
7. Figure 3. Possible effects of ABCA7 haploinsufficiency, based on experiments on cell- and animal models. NF-κB must be mentioned.
Thank you, we fixed this issue.
“Also, ABCA7 haploinsufficiency can inhibit the NF-κB pathway during the activation of microglia, leading reduced pro-inflammatory cytokine production.”
Reviewer 2 Report
Comments and Suggestions for Authors
The review article titled "Haploinsufficiency and Alzheimer’s Disease: The Possible Pathogenic and Protective Genetic Factors" is well-written and organized. It discusses the role of haploinsufficiency in Alzheimer's disease (AD), highlighting genetic factors that may either contribute to AD pathogenesis or offer neuroprotection through haploinsufficiency mechanisms. However, it needs to be revised according to the following comments:
1. The abstract needs to be concise and focused. Lines 24-30 resemble a results or discussion section, which is inappropriate for an abstract. It should instead emphasize the purpose of the review and outline the critical areas that require further exploration, presenting only the most relevant information.
2. Lines 47-50 state 'more recently' regarding frontotemporal dementia, but the cited references (1998, 2006) are outdated. Consider using more current references or adjusting the phrasing for accuracy.
3. In general, the authors should carefully review and standardize the use of abbreviations throughout the manuscript to ensure clarity and consistency. They should also address any typographical errors to enhance readability. Moreover, adding a dedicated section on conclusions and future perspectives would strengthen the review by summarizing key insights and suggesting directions for future research.
Author Response
Reviewer 2
The review article titled "Haploinsufficiency and Alzheimer’s Disease: The Possible Pathogenic and Protective Genetic Factors" is well-written and organized. It discusses the role of haploinsufficiency in Alzheimer's disease (AD), highlighting genetic factors that may either contribute to AD pathogenesis or offer neuroprotection through haploinsufficiency mechanisms.
Thank you for the positive review and encouraging comments. We will revise the manuscript, according to your suggestions.
However, it needs to be revised according to the following comments:
1. The abstract needs to be concise and focused. Lines 24-30 resemble a results or discussion section, which is inappropriate for an abstract. It should instead emphasize the purpose of the review and outline the critical areas that require further exploration, presenting only the most relevant information.
Thank you, we tried to fix this issue.
“Animal studies examining haploinsufficient AD risk genes, such as Vacuolar protein sorting-associated protein 35 (VPS35), sirtuin-3 (SIRT3), and PICALM, have shown that their knockout can exacerbate neurodegenerative processes by promoting amyloid production, accumulation, and inflammation. Conversely, haploinsufficiency in APOE, Beta-secretase 1 (BACE1), and Transmembrane protein 59 (TMEM59) has been reported to confer neuroprotection by potentially slowing amyloid deposition and reducing microglial activation. Given its implications for other neurodegenerative diseases, the role of haploinsufficiency in AD requires further exploration.​ Modeling the mechanisms of gene knockout and monitoring their expression patterns is a promising approach to uncover AD-related pathways. However, challenges such as identifying susceptible genes, gene-environment interactions, phenotypic variability, and biomarker analysis must be addressed. Enhancing model systems through humanized animal or cell models, utilizing advanced research technologies, and integrating multi-omics data will be crucial for understanding disease pathways and developing new therapeutic strategies."
2. Lines 47-50 state 'more recently' regarding frontotemporal dementia, but the cited references (1998, 2006) are outdated. Consider using more current references or adjusting the phrasing for accuracy.
Thank you, we added more recent references to the manuscript.
3. In general, the authors should carefully review and standardize the use of abbreviations throughout the manuscript to ensure clarity and consistency. They should also address any typographical errors to enhance readability. Moreover, adding a dedicated section on conclusions and future perspectives would strengthen the review by summarizing key insights and suggesting directions for future research.
Reviewer 1 had similar comments too. We fixed the abbreviations, and added a paragraph, discussing the limitations, and future insights.
“However, current studies on haploinsufficiency in diseases have several limitations. First, the majority of studies have been performed on model organisms (e.g., mice) or cell lines, which may not accurately reflect the effects of haploinsufficiency in humans. There may be differences in the gene expression patterns of the same mutations between animal models and humans. For example, the TREM2 Arg47His mutation resulted in reduced TREM2 expression in mice but not in humans [85, 166, 167, 168]. Another issue is that haploinsufficiency of certain genes can result in diverse phenotypes. Besides Alzheimer's disease (AD), truncation mutations in PSEN1 and PSEN2 have been associated with a wide range of disease phenotypes, such as frontotemporal dementia (FTD), acne inversa, and acute encephalopathy. This wide variety of phenotypes can complicate disease modeling [40-53]. Studying haploinsufficiency in humans also presents significant challenges due to the heterogeneity of the human population. Individuals from different populations often have diverse genetic backgrounds that can influence disease phenotypes [169]. Modifier genes and environmental factors may also affect the expression of haploinsufficient genes. In animal models, it can be difficult to model the impact of these genetic modifiers and gene-environment interactions concerning haploinsufficiency [170, 171, 172].
There are also technical challenges associated with studying haploinsufficiency. Detecting gene dosage can be difficult, and sensitive techniques (e.g., quantitative PCR or RNA sequencing) may be expensive and not accessible to everyone [173]. Furthermore, measuring the levels of certain proteomic markers, including membrane proteins (such as APP, PSENs or ABCA7) may be challenging. Membrane proteins are less abundant in bio-logical fluids, and difficult to be measured by conventional proteomic technologies. It may be challenging to isolate membrane proteins, since they can lose their stability. However, these membrane proteins can be cleaved by proteolytic enzymes. The fragment of mem-brane proteins, released to the plasma can be serve as alternative disease biomarkers [174, 175, 176]. Although CRISPR-Cas9 is a promising technique for modeling diseases, its results can be misleading due to the potential for off-target effects [177]. In the future, advanced gene-editing technologies may open new avenues for haploinsufficiency research in various diseases, including AD. Increased specificity and reduced off-target effects should enhance the ability to study the potential impacts of haploinsufficiency [177, 178, 179]. Also, the utilization of biomarkers for haploinsufficiency should be essential for the disease diagnosis and in understanding the possible disease associated pathways [180]. Multi-omics approaches (including genomics, transcriptomics, proteomics, and metabolomics) may also be useful for discovering how altered dosage influences AD-related pathways [181]. The development of humanized animal models could yield more accurate insights into disease modeling [182].’